# EVALUATING NATURAL LANGUAGE PROCESSING MODELS WITH GENERALIZATION METRICS THAT DO NOT NEED ACCESS TO ANY TRAINING OR TESTING DATA

## ABSTRACT

The search for effective and robust metrics has been the focus of recent theoretical and empirical work on generalization of deep neural networks (NNs). In this paper, we discuss the performance of natural language processing (NLP) models, and we evaluate various existing and novel generalization metrics. Compared to prior studies, we (i) focus on NLP instead of computer vision (CV), (ii) focus on generalization metrics that predict test error instead of the generalization gap, (iii) focus on generalization metrics that do not need the access to data, and (iv) focus on the heavy-tail (HT) phenomenon that has received comparatively less attention in the study of deep neural networks. We extend recent HT-based work which focuses on power law (PL) distributions, and we study exponential (EXP) and exponentially truncated power law (E-TPL) fitting to the empirical spectral densities (ESDs) of weight matrices. Our empirical studies are carried on (i) hundreds of Transformers trained in different settings, in which we systematically vary the amount of data, the model size and the optimization hyperparameters, (ii) a total of 51 pretrained Transformers from eight families of Huggingface NLP models, including BERT, GPT2, ALBERT, etc., and (iii) a total of 28 existing and novel generalization metrics. From our detailed empirical analyses, we show that *shape metrics*, or the metrics obtained from fitting the shape of the ESDs, perform uniformly better at predicting generalization performance than *scale metrics* commonly studied in the literature, as measured by the *average* rank correlations with the generalization performance for all of our experiments. We also show that among the three HT distributions considered in our paper, the E-TPL fitting of ESDs performs the most robustly when the models are trained in experimental settings, while the PL fitting achieves the best performance on well-trained Huggingface models, and that both E-TPL and PL metrics (which are both shape metrics) outperform scale metrics.

## 1 INTRODUCTION

Recent years have seen a wide array of large-scale empirical studies on the various metrics used to quantify generalization (Dziugaite et al., 2020; Jiang et al., 2019; Martin & Mahoney, 2021a; Martin et al., 2021). On the one hand, theory-driven metrics have the potential to reveal more information than test error, bringing us one step closer to unpacking the black box of deep NNs (Frankle & Carbin, 2018; Nakkiran et al., 2019; Zhang et al., 2021). On the other hand, a wide variety of generalization metrics have been applied to predict the *quality* of pretrained models (Martin & Mahoney, 2019; Martin et al., 2021), design effective training procedures (Foret et al., 2020; Izmailov et al., 2018), improve network efficiency (Chen et al., 2020; Dong et al., 2019), quantify network robustness (Tanay & Griffin, 2016; Yang et al., 2020), improve ensemble learning techniques (Fort et al., 2019; Garipov et al., 2018), analyze and improve large-scale machine learning contests (Martin & Mahoney, 2021a), and so on.

Despite advances in the study of generalization, however, several recent papers point out the deficiencies of many of these "fantastic" generalization metrics. These include a lack of "robustness" to the changes of environmental hyperparameters (Dziugaite et al., 2020; Jiang et al., 2019) (such as data, network architecture and training schemes), or the Simpson's paradox that generalization metrics perform differently (i.e., predict opposite trends) when applied to each sub-part of a collection

of learning models or to the holistic study (Martin & Mahoney, 2021a). Another drawback is the over-reliance on experiments with CV models, which are relatively well-explored, and which are not representative of many other application areas. Despite a few counterexamples (Martin et al., 2021; Nakkiran et al., 2019; Yang et al., 2021), systematic studies of generalization in other fields, such as NLP, are largely missing.

**Generalization metrics for NLP.** The objective of this paper is to provide a systematic study of generalization metrics in NLP, addressing several deficiencies in prior studies (Dziugaite et al., 2020; Jiang et al., 2019; Martin et al., 2021). Compared to CV, predicting generalization in NLP has several important differences that require careful consideration. The training data from standard CV benchmarks can often be easily obtained, while NLP pretraining datasets are typically web-scale and are challenging to access. Therefore, generalization metrics that can measure the quality of learning models *without access to data* are ideal for NLP. Indeed, recent work has demonstrated that access to training or testing data is *not* necessary for assessing the model quality of learning models (Martin et al., 2021), though these have yet to be evaluated at scale in the NLP domain. Furthermore, it is typically infeasible to train NLP models to interpolate the (frequently large) training set. This becomes an issue when applying most existing generalization metrics as they often estimate the *generalization gap* (i.e., the difference between training and test performance) rather than the test error itself. Metrics that focus on predicting the generalization gap include most of the well-known metrics in CV, such as those based on the PAC-Bayesian framework (McAllester, 1999; Neyshabur et al., 2018) and margins (Bartlett et al., 2017; Jiang et al., 2018; Pitas et al., 2017).

To illustrate the issue, consider the problem of model selection between two models (Jiang et al., 2020; Martin & Mahoney, 2021a).Suppose we are given two classification models. Then even if we have i) access to both models' training errors, and ii) a metric which is guaranteed to perfectly rank correlate with the generalization gap, then we still cannot determine which model as smaller test error. This means that, if our objective is to construct a metric that correctly predicts which model has lower test error, rank correlation with the generalization gap is not sufficient. In this paper, we aim to study how generalization metrics correlate with *model quality*, for which we use test error as a close approximation. As we will demonstrate (in Figure 4), rank correlation with the generalization gap indeed does not imply rank correlation with model quality in practice, and in fact often orders models in the opposite order of their test errors. From a practical point of view, for NLP tasks, we prefer generalization metrics that can directly predict trends in test error (or similar evaluation metrics in NLP, such as the test BLEU score (Papineni et al., 2002)) rather than trends in the generalization gap.

Naturally, we cannot expect a metric to be universally correlated with test error if evaluating the metric does not need data. However, within certain classes of models (e.g., stages of training in one model or across pre-trained models), they may be effective at diagnosing model quality. With these objectives in mind, among the generalization metrics in the literature, we take particular interest in those derived from the heavy-tail self regularization (HT-SR) theory (Martin & Mahoney, 2019, 2021b), which (i) predicts test error directly instead of the generalization gap and (ii) does not require access to training (or testing) data.

**HT-SR theory.** The core principle of HT-SR theory is that HT structures arise naturally in the ESDs of the weight matrices [1] as the result of extracting various correlations in data during optimization (Martin & Mahoney, 2019, 2021a,b; Martin et al., 2021). Its primary practical consequence is that by estimating the PL coefficient from the ESDs (requiring only weights), one can predict model quality, as smaller coefficients are reported to correspond to higher test accuracy. However, these estimators can be unstable, and so one must be careful not to rely on them alone. The quality of the PL fit itself should also point to similar conclusions (Martin & Mahoney, 2021b), which can be a sanity check.

The principles of HT-SR theory extend beyond fitting the PL coefficient, however, as ESDs can take many forms. To this end, we study three different types of distributions to fit to the ESDs of weight matrices, including power laws (PL) in Eqn. (1), exponentially truncated power laws (E-TPL) in Eqn. (2), and exponential laws (EXP) in Eqn. (3). These are all commonly considered families of distributions in classical studies of PL (Clauset et al., 2009), and it is often hard in practice to predict which family fits data the best (as we show in this paper, this is true for deep NNs especially).

---

[1]The ESD of a weight matrix $\mathbf{W}$ refers to the empirical density of the eigenvalues of the squared weight matrix $\mathbf{W}^{\top}\mathbf{W}$. See "Preliminary of ESDs of weight matrices" at the end of the Introduction.

Figure 1 shows examples of comparing different HT fittings on the same ESD. Following Martin & Mahoney (2021a), we refer to the various metrics derived from HT-SR as *shape metrics*.

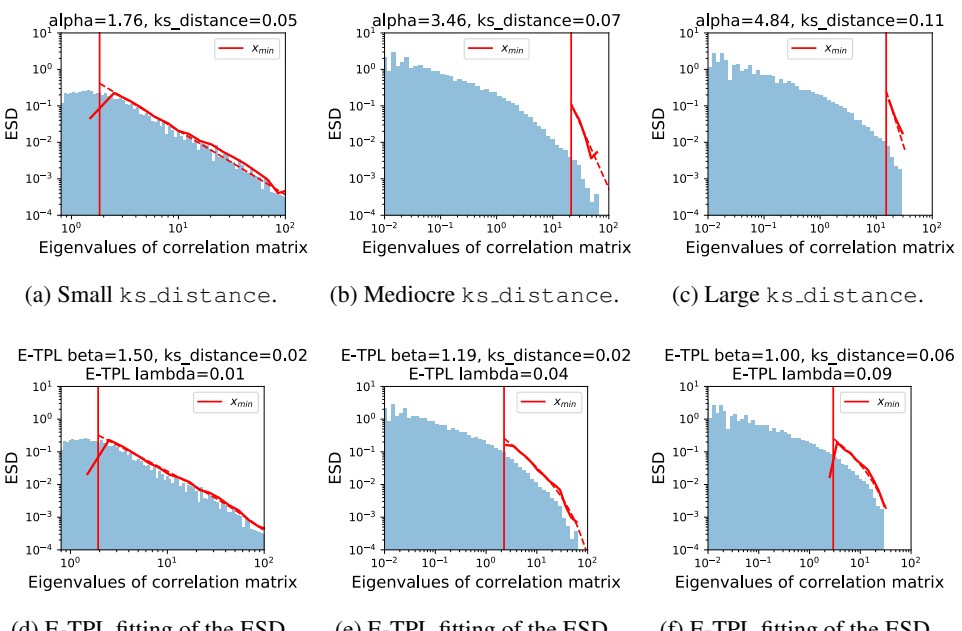

(a) Small ks_distance.  (b) Mediocre ks_distance.  (c) Large ks_distance.

(d) E-TPL fitting of the ESD.  (e) E-TPL fitting of the ESD.  (f) E-TPL fitting of the ESD.

Figure 1: Comparing PL and E-TPL fitting. **(First row).** Good, mediocre, and bad PL fittings measured by the ks_distance. **(Second row).** E-TPL fitting of the ESD on the same column. Blue histograms represent the ESDs. Solid vertical lines represent the lower threshold $x_{\min}$ of the PL distribution found by the fitting procedure. Solid curves represent ESDs truncated using $x_{\min}$, and dashed curves represent the fitted HT distributions.

**Contributions.** The following summarizes our main contributions.

- Deviating from prior work examining generalization metrics in CV (Dziugaite et al., 2020; Jiang et al., 2019), we provide the first systematic empirical study on various generalization metrics in NLP. Our detailed studies include the following:
  - We consider 360 transformers trained with varying hyperparameters, and eight families of pretrained SOTA transformers downloaded from Huggingface (Wolf et al., 2020), including BERT (Kenton & Toutanova, 2019), GPT2 (Radford et al., 2019), ALBERT (both v1 and v2) (Lan et al., 2019), etc.
  - measuring the correlation between 28 generalization metrics and the model quality (measured by test-time performance) over three different model classes: (i) models trained with the optimal hyperparameters; (ii) a single model at different stages of training; and (iii) a model trained with different hyperparameters (similar to Jiang et al. (2019)).
- We extend prior studies on HT-SR theory and investigate alternative models to fit heavy-tail/light-tail distributions. Our results show that E-TPL fits are comparatively robust alternatives to PL fits for predicting trends in test error on suboptimally-trained models.
- We find that, applied appropriately, HT-based shape metrics consistently perform better than scale metrics (or norm-based metrics) for predicting model quality.
- We provide results for data-dependent metrics motivated by margins and PAC-Bayesian bounds (Dziugaite et al., 2020; Jiang et al., 2019). While these metrics perform well in predicting the generalization gap, we show that none of them satisfactorily predicts test error directly.

**Preliminary of ESDs of weight matrices.** Consider a NN with $d$ layers and corresponding weight matrices $\mathbf{W}_1, \mathbf{W}_2,..., \mathbf{W}_d$. For each weight matrix $\mathbf{W}_i$ with shape $N \times M$, assume without loss of generality that $N \geq M$ (otherwise, consider $\mathbf{W}_i^\top$). We define the correlation matrix as $\mathbf{X}_i = \mathbf{W}_i^\top \mathbf{W}_i$, and denote the eigenvalues of $\mathbf{X}_i$ as $\{\lambda_j\}_{j=1}^M$, so that $\lambda_j = \sigma_j^2$, where $\{\sigma_j\}_{j=1}^M$ are the singular values of $\mathbf{W}_i$. Furthermore, we use $\lambda_{i,\max}$ to denote the maximum eigenvalue of the correlation matrix $\mathbf{X}_i$. The ESD of the weight matrix $\mathbf{W}_i$ refers to the empirical density of the

eigenvalues of $\mathbf{X}_i$, typically represented through a histogram. We let $p(x)$ denote the density function to fit the ESD taking values in the interval $(x_{\min}, x_{\max})$. For a power law, $p$ satisfies

$$p(x) \propto x^{-\alpha}, \quad x_{\min} < x < x_{\max}. \tag{1}$$

From Martin & Mahoney (2021a), $x_{\max}$ is chosen to be the maximum eigenvalue of the empirical correlation matrix. However, $x_{\min}$ is a variable to be optimized to improve the quality of PL fitting, and it is not equal to the minimum eigenvalue in general.

## 2 HEAVY-TAIL SELF-REGULARIZATION THEORY

Here, we provide a brief overview of the HT-SR theory, and discuss several metrics that can be derived from it. According to HT-SR theory, the ESDs of the weight matrices become more heavy-tailed during training as they become increasingly correlated. One can quantify the extent of these correlations by fitting a PL to the ESD of a weight matrix, for example, by using the open-source `WeightWatcher` tool (Martin et al., 2021). After computing the ESD of a weight matrix, we use the maximum likelihood estimate from Alstott et al. (2014) to fit the PL distribution, the specific form of which has been defined in (1). Let `PL_alpha` denote the PL coefficient averaged over layers; effectively the slope of the tail of the ESD of the pooled weights, on a log-log scale.

Correctly identifying and fitting PL distributions is well-known to be a challenge in practice. For example, a density that appears as a straight line on a log-log scale plot need not follow a power law, as there are many other distributions that could show a similar behavior, including lognormal and exponential-type distributions (Clauset et al., 2009). Nested distributions such as E-TPL, which combine the pure PL and other distributional assumptions, can often improve the quality of fitting (Alstott et al., 2014; Clauset et al., 2009). Therefore, in addition to the PL distribution defined in (1), we consider several other distribution classes from the literature.

- (`E_TPL_lambda` and `E_TPL_beta`) The ESDs are assumed to take a "nested" form.

$$p(x) \propto x^{-\beta} \exp(-\lambda x), \, x_{\min} < x < x_{\max}. \tag{2}$$

  After fitting the E-TPL, we call the exponential truncation coefficient $\lambda$ the `E_TPL_lambda` metric, and we call the PL coefficient the `E_TPL_beta` metric.

- (`EXP_lambda`). The ESDs are assumed to take the following form.

$$p(x) \propto \exp(-\lambda x), \, x_{\min} < x < x_{\max}. \tag{3}$$

  After fitting the EXP, we call the exponential coefficient $\lambda$ the `EXP_lambda` metric.

For more details of the various metrics considered in this paper, see Table 1. All of the metrics derived from HT-SR do *not* require access to data, and they are relatively cheap to compute. Our primary comparisons are between shape metrics (derived from HT-SR), and scale metrics (mostly norm-based). For the precise definitions of these metrics, see Appendix A.

**Issues of PL fitting.** It is well-known that subtle issues can arise when fitting the ESDs (Alstott et al., 2014; Clauset et al., 2009; Martin & Mahoney, 2017, 2021a). To best mitigate these issues in PL fits, we adopt the fitting strategies used in WeightWatcher (Martin & Mahoney, 2017). For example, as in Clauset et al. (2009), it is common to choose the lower threshold $x_{\min}$ which coincides with the best quality fit under the Kolmogorov–Smirnoff statistic (referred to as `PL_ks_distance` for PL and `E_TPL_ks_distance` for E-TPL in the sequel; see Eqn. (12)). However, this method is time-consuming, especially for E-TPL as there are two parameters to fit. Instead, we adopt the *fix-finger method* (see `WeightWatcher`) which selects $x_{\min}$ as the peak of the ESD when fitting E-TPLs. More than a simple speed improvement, we find this method also yields more stable results.

**Comparing PL and E-TPL fitting.** Referring to Figure 1, we now discuss how E-TPL could partially address these fitting issues. On the first row of Figure 1, we show three typical cases of PL fitting. In Figure 1a, the log-log scale reveals a "linear region" of the histogram, which the PL fitting correctly locates. The quality of fit, measured by the `ks_distance`, is within a typical range, as reported in Table 5 of Martin & Mahoney (2021b). In Figure 1b and Figure 1c, the ESDs do not exhibit a clear linear region on the log-log scale. Following Martin & Mahoney (2021b), it is ill-advised to consider metrics derived from a PL fit in these scenarios. In practice, this typically

| Name | Eqn | Ref | Need initial weights? | Scale or shape | Need data? | Need gpu? | Predicting model quality or generalization gap? |
|---|---|---|---|---|---|---|---|
| param_norm | (4) | Jiang et al. (2019) | No | Scale | No | No | Generalization gap |
| fro_dist | (5) | Jiang et al. (2019) | Yes | Scale | No | No | Generalization gap |
| log_norm | (6) | Martin & Mahoney (2021b) | No | Scale | No | No | Generalization gap |
| log_spectral_norm | (7) | Martin & Mahoney (2021a) | No | Scale | No | No | Generalization gap |
| dist_spec_int | (8) | Jiang et al. (2019) | Yes | Scale | No | No | Generalization gap |
| path_norm | (9) | Neyshabur et al. (2015) | No | Scale | No | No | Generalization gap |
| mp_softrank | (10) | Martin & Mahoney (2021b) | No | Scale/Shape | No | No | Model quality |
| stable_rank | (11) | Martin & Mahoney (2021b) | No | Scale/Shape | No | No | Model quality |
| PL_alpha | (1) | Martin & Mahoney (2021b) | No | Shape | No | No | Model quality |
| E_TPL_beta | (2) | This paper WeightWatcher | No | Shape | No | No | Model quality |
| E_TPL_lambda | (2) | This paper WeightWatcher | No | Shape | No | No | Model quality |
| EXP_lambda | (3) | WeightWatcher | No | Shape | No | No | Model quality |
| PL_ks_distance | (12) | Martin & Mahoney (2021b) | No | Shape | No | No | Model quality |
| E_TPL_ks_distance | (12) | This paper Martin & Mahoney (2021b) | No | Shape | No | No | Model quality |
| alpha_weighted | (13) | Martin & Mahoney (2021b) | No | Hybrid | No | No | Model quality |
| log_alpha_norm | (14) | Martin & Mahoney (2021a) | No | Hybrid | No | No | Model quality |
| inverse_margin | (17) | Jiang et al. (2019) | No | Scale | Yes | Maybe | Generalization gap |
| log_prod_of_spec_over_margin | (18) | Bartlett et al. (2017) Pitas et al. (2017) | No | Scale | Yes | Maybe | Generalization gap |
| log_sum_of_spec_over_margin | (19) | Bartlett et al. (2017) Pitas et al. (2017) | No | Scale | Yes | Maybe | Generalization gap |
| log_prod_of_fro_over_margin | (20) | Bartlett et al. (2017) Pitas et al. (2017) | No | Scale | Yes | Maybe | Generalization gap |
| log_sum_of_fro_over_margin | (21) | Bartlett et al. (2017) Pitas et al. (2017) | No | Scale | Yes | Maybe | Generalization gap |
| path_norm_over_margin | (22) | Neyshabur et al. (2015) | No | Scale | Yes | Maybe | Generalization gap |
| pacbayes_init | (25) | Neyshabur et al. (2017) | Yes | Scale | Yes | Yes | Generalization gap |
| pacbayes_orig | (26) | Neyshabur et al. (2017) | No | Scale | Yes | Yes | Generalization gap |
| pacbayes_flatness | (27) | Neyshabur et al. (2017) | No | Scale | Yes | Yes | Generalization gap |
| pacbayes_mag_init | (28) | Jiang et al. (2019) | Yes | Scale | Yes | Yes | Generalization gap |
| pacbayes_mag_orig | (29) | Jiang et al. (2019) | No | Scale | Yes | Yes | Generalization gap |
| pacbayes_mag_flatness | (30) | Jiang et al. (2019) | No | Scale | Yes | Yes | Generalization gap |

Table 1: Overview of the generalization metrics considered. We focus on the *shape* metrics derived from the ESDs of weight matrices. See Appendix A for the details of these metrics.

occurs when PL_alpha > 4 (e.g., see Figure 1c). On the other hand, in these two cases, the corresponding E-TPL fits (shown on the second row in Figure 1) still closely match the empirical density function (see Figure 1e and Figure 1f), and the ks_distance on the second row using a E-TPL fit is smaller than that for the PL fit on the first row, even when the fit on the second row clearly covers a larger part of the ESD. In these two cases, the E_TPL_lambda plays a similar role as the PL_alpha in PL fitting, and provides an effective alternative when the ESD does not exhibit a proper PL.

Between these three PL and E-TPL fittings, we would like to point out that the important thing in HT-SR is not the PL fitting *per se* but that the spectral distributions exhibit HT or other non-standard shapes. The particular forms of the distributions fit here simply constitute different ways to quantify this property in practice. These details, such as selecting the most appropriate distributional assumptions, clearly matter if we would like to engineer the tools of HT analysis to effectively measure the ground truth. However, the primary concern in predicting generalization is to measure the shape information, and the shape information is independent of the fitting procedure, although better fitting procedures may capture the shape information better.

## 3 EMPIRICAL RESULTS

### 3.1 EXPERIMENTAL SETUP

**Dataset.** We consider the WMT14 German to English (DE-EN) dataset (Bojar et al., 2014), commonly used as a benchmark for neural machine translation (Edunov et al., 2018; Ott et al., 2018; Shen et al., 2020; Vaswani et al., 2017). WMT14 consists of 4.5 million sentence pairs for training.

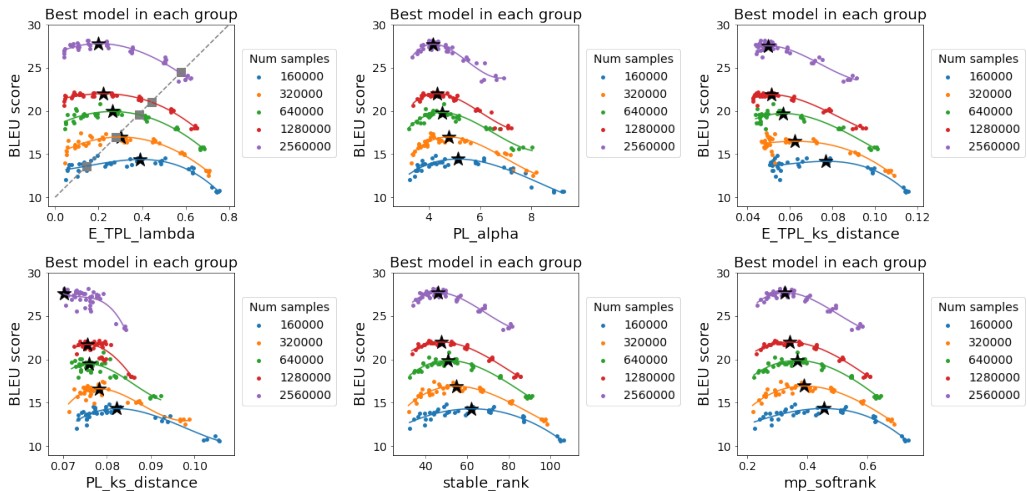

Figure 2: BLEU-score vs. six shape metrics for 200 Transformers trained on WMT14 with varying hyperparameters. HT-SR theory applies for optimally-tuned models (black stars), that is, models that have better BLEU scores exhibit heavier-tailed ESDs. For suboptimal models, the HT-SR metrics can be anti-correlated with model quality, see e.g. the grey dotted line in the first subfigure.

**Hyperparameters.** To capture the relationship between the generalization metrics and model quality in a number of different settings, we vary several hyperparameters: the number of samples (either 160K, 320K, 640K, 1.28M, 2.56M samples), the initial learning rate during training (across eight different rates), the model width (embedding dimension either 256, 384, 512, 768, or 1024), and the model depth ({4, 5, 6, 7, 8}-layer transformers). We also construct a high-dimensional grid of different hyperparameters $\Theta = \{(\theta_1, \ldots, \theta_K) : \theta_1 \in \Theta_1, \ldots, \theta_K \in \Theta_K\}$, so that we can compare models when one of the hyperparameters is varied. Two separate high-dimensional grids with dimension $K = 3$ are considered: (1) sample×learning rate×width; (2) sample×learning rate×depth. Each grid contains 5×8×5=200 of these training settings. In total, there are 360 trained models because the two high-dimensional grids overlap each other, and 40 models belong to both grids. We will consider three subtasks to evaluate the considered generalization metrics.

**Task one, correlation evaluated on optimally trained models.** In the first task (Section 3.2.1), we study the relationship between model quality and generalization metrics on models trained with the optimal choice of hyperparameters. This task mimics the grid-search method often employed in large-scale (pre)training tasks.

**Task two, correlation in time.** In the second task (Section 3.2.2), we track BLEU score and generalization metrics during training, assessing time-wise correlation to model quality. This task has been considered in the literature (Bartlett et al., 2017), and from a practical point of view, capturing the time-wise dependence during training could potentially lead to better ways of early stopping or regularizing the model.

**Task three, correlation when hyperparameters are varied.** In the third task , we study the relationship between the model quality and the generalization metrics when a single hyperparameter is varied. Metrics that achieve a high (rank) correlation for all the hyperparameters are good candidates for model selection. Constrained by space, we discuss this result in details in Appendix B.

**Training and model setup.** For the details of the training settings, see Appendix C.

## 3.2 EVALUATING THE METRICS ON TRANSFORMERS TRAINED IN DIFFERENT SETTINGS

In this subsection, we study 28 generalization metrics (with details provided in Table 1) and examine their correlations with BLEU score (Papineni et al., 2002), the most commonly used metric to evaluate

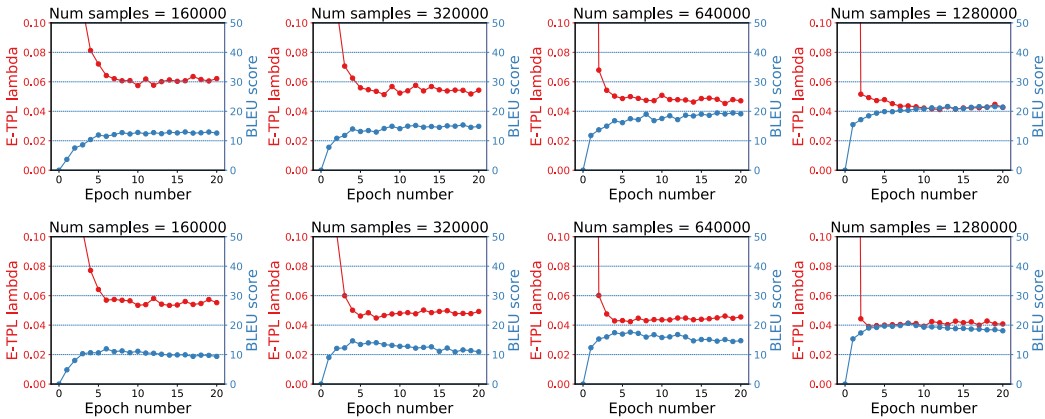

Figure 3: `E_TPL_lambda` closely tracks the BLEU score, i.e., BLEU score increases when the `E_TPL_lambda` drops. Results are shown for Transformers trained on WMT14 with different number of samples. **(First row).** Training with dropout 0.1. **(Second row).** Training without dropout.

machine translation [2]. We also consider correlation between these metrics and the generalization gap, defined as the BLEU score for training data subtracted by the BLEU score for test data.

### 3.2.1 TASK ONE: EVALUATING CORRELATIONS ON OPTIMALLY TRAINED MODELS ONLY

Here, we group models using the number of training samples, and select the best model from each group when the model depth and the learning rate are varied. In Figure 2, each curve represents a group of models trained with a certain number of training samples. The black star on each curve represents the model trained with the optimal choice of hyperparameters (learning rate and depth in our setting), obtained by searching for the optimum on a third-order polynomial fit of each curve. From Figure 2, we see that the shape metrics correctly predict the model quality for models trained with the optimal hyperparameters, i.e., the BLEU scores should be higher when the metric values are smaller.

### 3.2.2 TASK TWO: TIME-WISE CORRELATIONS AND RANK CORRELATION RESULTS

In this subsection, we study time-wise correlation between our chosen metrics and the BLEU scores.

**`E_TPL_lambda` tracks the BLEU score.** As a warm-up, we consider how well the `E_TPL_lambda` metric defined in (2) tracks the BLEU score (recalling that `E_TPL_lambda` assumes the ESDs follow E-TPLs). We use training with and without dropout to study the effect of training schemes, and we consider different quantities of data to test robustness in the dataset. In Figure 3, the first row considers models trained with dropout, while the second row considers models trained without dropout. The multiple columns track `E_TPL_lambda` and the BLEU score throughout training for different amounts of data. We can see that `E_TPL_lambda` not only successfully tracks BLEU scores but also differentiates underfitting (first row, with dropout) from overfitting (second row, without dropout) in this experiment.

**Shape metrics predict model quality, while scale metrics predict the generalization gap.** Now we consider the rank correlations between our chosen metrics and the test BLEU score. The rank correlations are evaluated across training, i.e., for each of the 360 settings of the hyperparameters, we calculate the Spearman's rank correlation between BLEU scores and the values of each generalization metric over all epochs. The summarized results are presented in Figure 4a. A positive Spearman's rank correlation (with BLEU) suggests that the generalization metric is useful in tracking BLEU during training. A negative Spearman's rank correlation, on the other hand, implies that the metric

---

[2]Several empirical metrics have been designed to measure the quality of text generation, such as BERTScore (Zhang et al., 2019) and BARTScore (Yuan et al., 2021). Our work is different because we do not need any data, and we do model selection using the weight matrices only. BERTScore and BARTScore evaluate the text quality, and thus they need source or reference texts generated by humans. These metrics can serve as alternatives to BLEU, which is viewed as ground truth in our work.

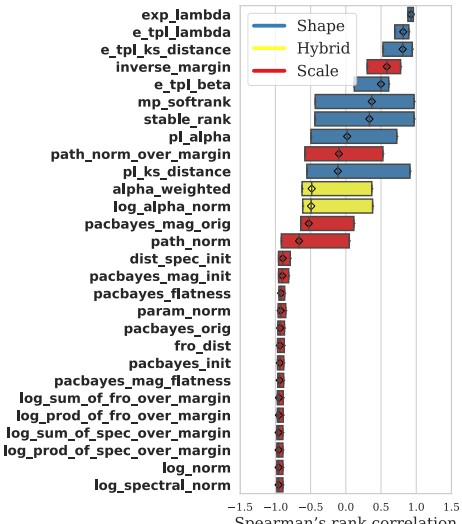
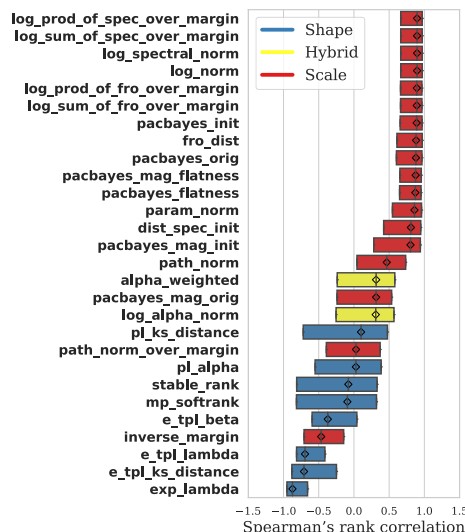

(a) **Correlations with model quality.** Spearman's rank correlation between various generalization metrics and BLEU.

(b) **Correlations with generalization gap.** Spearman's rank correlation between various generalization metrics and the generalization gap.

Figure 4: Comparing multiple generalization metrics for predicting BLEU score (on the left) or the generalization gap (on the right). Lines on each box delineate the 25/50/75 percentiles of the rank correlations in 360 different settings (including different amount of data, different network depths, different network widths, and different initial learning rates).

often gives the incorrect prediction. In Figure 4a, we use the average rank correlations for all settings to study the effectiveness of each metric, and present 25% quantile rank correlations to indicate robustness across runs.

In Figure 4a, we find shape metrics, such as EXP_lambda, E_TPL_lambda, E_TPL_ks_distance, and E_TPL_beta, exhibit some of the highest rank correlations with BLEU score. The EXP_lambda metric, which assumes a EXP distribution on the ESDs, achieves the highest median rank correlation, while the E_TPL_lambda metric, which assumes a E-TPL distribution on the ESDs, achieves the second highest. We discuss the inverse_margin metric in Appendix D.

In Figure 4b, we plot the rank correlations to the generalization gap across our chosen metrics. While it is encouraging that most existing generalization metrics yield correct predictions, as previously discussed, correct predictions of the generalization gap do *not* imply accurate predictions on the best-performing models here.

**Details of the rank correlation calculations.** When calculating the rank correlation with the test accuracy, we associate a negative sign to all the generalization metrics, i.e., a positive rank correlation in Figure 4a means that a generalization metric is negatively correlated with the BLEU score. We use this procedure to follow the conventional wisdom that a smaller value of the complexity metric leads to better generalization. On the other hand, for Figure 4b, a positive rank correlation means that the metric is positively correlated with the generalization gap. Thus, for both Figure 4a and 4b, a strong positive correlation corresponds to the expected trend.

**Rank-correlation results when varying a hyperparameter.** We also assess whether the generalization metrics can predict trends in BLEU score when a single hyperparameter is changed (Jiang et al., 2019). Our findings are shown in Appendix B. Again, shape metrics have better rank correlations with model quality, while scale metrics are better correlated with the generalization gap.

**Corroborating results.** We extend our empirical evaluations to other datasets and evaluation methods. In Section E.1, we consider three other language processing tasks trained with different Transformers. In Section E.2, we evaluate correlations using Kendall's tau instead of Spearman's rank correlation.

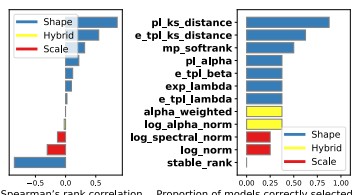 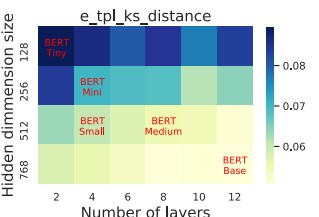 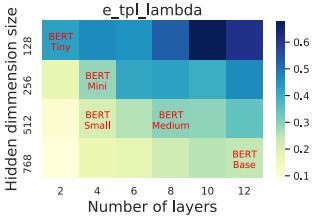

(a) Model selection on Huggingface Transformers. Metrics on the left and right are aligned.

(b) `E_TPL_ks_distance` evaluated on BERT models of different size.

(c) `E_TPL_lambda` evaluated on BERT models of different size.

Figure 5: Generalization metrics evaluated on pretrained Transformers. (a) Model selection results on eight Huggingface Transformer model series: BERT, GPT2, ALBERTv1, ALBERTv2, T5, DialoGPT, FlauBERT, Funnel Transformer. Left shows the rank correlation averaged over different Transformers. Right shows the proportion of the best Transformers correctly selected using different metrics. Shape metrics outperform scale metric only except `stable_rank` which is strongly influenced by the matrix size. (b and c) Evaluating two metrics on the "Smaller BERT" series. While `E_TPL_ks_distance` predicts the correct trends, `E_TPL_lambda` shows the reversed trends with depth.

## 3.3 EVALUATING THE GENERALIZATION METRICS USING HUGGINGFACE TRANSFORMERS

Finally, we evaluate the data-free generalization metrics on pretrained Transformers. Eight series of models downloaded from Huggingface (Wolf et al., 2020) are considered—see Table 2 for details. We also include 24 BERT models from the "Smaller BERT" series (Turc et al., 2019) produced from a "pretrained distillation" pipeline that combines masked language modeling pretraining (Kenton & Toutanova, 2019) and knowledge distillation from a single BERT teacher model. In total, there are 51 pretrained Transformers.

| Model series | Models |
|---|---|
| BERT (Kenton & Toutanova, 2019) | BERT-Tiny, BERT-Mini, BERT-Small, BERT-Medium, BERT-Base, BERT-Large |
| Smaller BERT (Turc et al., 2019) | 24 smaller BERT models (English only, uncased, trained with WordPiece masking) |
| GPT2 (Radford et al., 2019) | GPT2, GPT2-medium, GPT2-large, GPT2-xl |
| ALBERTv1 (Lan et al., 2019) | ALBERT-base-v1, ALBERT-large-v1, ALBERT-xlarge-v1, ALBERT-xxlarge-v1 |
| ALBERTv2 (Lan et al., 2019) | ALBERT-base-v2, ALBERT-large-v2, ALBERT-xlarge-v2, ALBERT-xxlarge-v2 |
| T5 (Raffel et al., 2020) | T5-small, T5-base, T5-large |
| DialoGPT (Zhang et al., 2020) | DialoGPT-small, DialoGPT-medium, DialoGPT-large |
| FlauBERT (Le et al., 2020) | FlauBERT_small_cased, FlauBERT_base_cased, FlauBERT_large_cased |
| Funnel Transformer (Dai et al., 2020) | FunnelModel-small, FunnelModel-medium, FunnelModel-intermediate FunnelModel-large, FunnelModel-xlarge |

Table 2: Pretrained Transformers considered in this paper.

We report rank-correlations averaged over these 8 model series in Figure 5a (left subplot), i.e., larger/deeper models should have smaller generalization metric values. Again, we find that the shape metrics outperform scale metrics (except for `stable_rank`, which is strongly influenced by the size of the weight matrix). The hybrid models achieve performance in-between the shape and scale metrics. In Figure 5a (right subplot), we compare different metrics in their ability to select the best model. That is, we report for each metric the proportion that the best model is selected from one model series when this metric is used as the model selection criterion. Note that the rankings of metrics on the two subplots in Figure 5a are the same.

From Figure 5a, we can see that, while the shape metrics perform better than scale metrics, none show a particularly strong rank correlation. To understand this, we examine the "Smaller BERT" series (Turc et al., 2019), which contains a more fine-grained structure of different model sizes. Specifically, these models are arranged in a 4-by-6 grid, where 6 represents {2,4,6,8,10,12} transformer layers and 4 means different hidden embedding sizes {128,256,512,768}. From Figure 5b, we see that the `E_TPL_ks_distance` correctly predicts the trend that wider and deeper models perform better. On the other hand, from Figure 5c, `E_TPL_lambda` correctly predicts that wider models are better, but incorrectly predicts that shallower models are better (yet another form of Simpson's paradox in a data set of neural network model quality; see also Martin & Mahoney (2021a)).

Another curious observation from Figure 5a is that, for the pretrained transformers, PL metrics, such as `PL_alpha` and `PL_ks_distance`, outperform E-TPL metrics, such as `E_TPL_lambda`,

`E_TPL_beta` and `E_TPL_ks_distance`. This phenomenon may seem surprising as one may expect E-TPL fits to be more flexible than PL fits. These pretrained models are likely trained with much larger datasets and over many more epochs than the models we have otherwise considered. Here, PLs appear to provide a more natural fit. This is further evidence that HT-SR theory is particularly well-suited for evaluating the quality of relatively high-quality models.

## 4 CONCLUSION

After conducting large-scale empirical studies on a variety of metrics, we find that shape metrics derived from HT-SR theory are more effective for model selection and model evaluation, in particular for evaluating models even without access to training or testing data. Poor correlations between existing generalization metrics and test-time performance have been reported in prior work (Dziugaite et al., 2020; Jiang et al., 2019; Nagarajan & Kolter, 2019). Rather than providing a "lump sum" to rank existing and novel generalization metrics (Figure 4), we evaluated these metrics in several ways: quantifying correlations only on optimally-trained models (Figure 2); examining the time-wise correlation during training (Figure 3); differentiating between the correlation with test accuracy versus generalization gap (Figure 4); evaluating these metrics on pretrained Transformer models where we do not have any control over the training process (Figure 5); and thoroughly investigating the rich correlational structures when different hyperparameters are varied (Figures 6 to 11 in the appendix). By thorough empirical investigations, we show that shape metrics perform consistently better than scale metrics in model selection—they correlate primarily with test accuracy instead of generalization gap, and they display better correlations with models' test performance. These metrics from HT-SR theory provide value to practitioners, allowing one to assess pretrained NLP models without training or testing data, even when their corresponding training loss is not small. Also, there are many large linear layers for Transformer models typically used in modern NLP tasks, which allows for greater accuracy in the PL estimators. That being said, further exploration of the Transformer architecture can lead to improved metrics on NLP models, complementing existing metrics designed explicitly for convolutional layers (Long & Sedghi, 2019; Sedghi et al., 2018). We expect our current and future studies to be relevant and useful for improving existing generalization metrics in NLP moving forward.

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
