# OpenReview forum: "Evaluating natural language processing models with generalization metrics that do not need access to any training or testing data"
_ICLR.cc/2023/Conference — Submitted to ICLR 2023_

### Official Review · Reviewer_oVEt · 2022-10-23

**Confidence:** 3
**Correctness:** 3
**Technical Novelty And Significance:** 2
**Empirical Novelty And Significance:** 3
**Recommendation:** 6

**Clarity, Quality, Novelty And Reproducibility:**

The work is generally clear and well-written (with a few small exceptions), and makes a significant contribution to the field. It is not wholly original as it does not propose any technically novel methods, but the scope and thoroughness of the study nevertheless constitute valuable and novel contributions. My only qualm with this is the limitation of focus to only a single dataset, which may make the results of questionable value outside the scope of that one dataset and task.

**Strength And Weaknesses:**

**Strengths**

The idea of producing a comprehensive study of generalization metrics for NLP models is an interesting and valuable contribution, and the proposed metric does indeed perform well in the thorough empirical studies they performed. The authors tested a wide variety of hyperparameter settings and convincingly showed strong performance on predicting generalization ability. The analysis of existing pretrained models also led to some interesting results, and considered a wide variety of models from the literature with different variants.

**Weaknesses**

My primary criticism of this paper is that almost all this analysis was performed only on a single dataset:  the WMT '14 english to german translation dataset. As such, it is difficult to be entirely confident that the results shown in the paper are truly general, and not specific to either this dataset or machine translation models on the whole. An analysis of multiple datasets with diverse task settings and dataset sizes that could show consistent results would be significantly more convincing. I certainly understand that this must already have been an enormous computational undertaking, but surely some of the resources used in the broad hyperparameter search might be better spent on replicating these results with other datasets. This is particularly critical given that this paper's primary contribution seems to be the broad scope of empirical analysis, rather than technical novelty.

My one other criticism is that I believe it is extremely important that papers relying on large-scale experiments accurately report the computational cost of their experiments and particularly the approximate greenhouse gas emissions of said experiments (there are a number of calculators online for estimating this). It is important for us to be cognizant of the environmental implications of our research and I believe transparency about the environmental cost of our experiments is ethically significant.

Questions/Minor errors:
1) I don’t believe ESD is defined anywhere in the introduction of the paper outside of the abstract
2) Figure 2 is a little bit unclear. You state that "the shape metrics always correctly predict the model quality, i.e., the BLEU scores should be higher when the metric values are smaller". Is this true only for the black starred models (the optimal configurations)? The wording makes it seem like this holds more generally, but this is not visually obvious and is contradicted by the caption.
3) In Figure 4 the 'scale' metrics seem near-perfectly anticorrelated with model quality. Shouldn't that mean these metrics are actually *more* predictive of model quality than the shape metrics, as a perfect negative indicator should be just as valuable as a perfect positive one?


**Summary Of The Paper:**

This paper proposes a metric to measure generalization ability for transformer models that is not data-dependent. Their proposed metric is based entirely on a spectral analysis of the weight matrices, and the authors seek to show that this metric correlates directly with test accuracy (as a proxy for model generalization) rather than the 'generalization gap' considered by other works. They demonstrate the effectiveness of their generalization metric on transformers trained with a wide variety of hyperparameter settings on WMT '14, as well as analyzing its predictions on a number of pretrained models from the literature such as BERT, GPT-2 and T5.

**Summary Of The Review:**

Overall I think the paper makes a valuable contribution in thoroughly cataloguing generalization metrics for NLP models and demonstrating the effectiveness of HT-SR methods at this task in a data-agnostic way. If these results had been demonstrated on a broader array of natural language tasks and datasets I would have no qualms about recommending it be accepted. As is, I believe this is a significant enough limitation for a work whose primary novelty is in the scope of its empirical studies that I view it as more of a borderline paper.

---

> ### Author Response · Authors · 2022-11-11
> **Response**
>
> **A single dataset.**
>
> See the comment on new experiments.
>
> **Greenhouse gas emissions.**
>
> The overall training cost is 7301.66 GPU hours. We use GPU nodes with TITAN RTX for our training. The overall carbon emission depends on carbon efficiency. Using the default values from the following website, total emissions are estimated to be 883.21 kgCO2eq.
> https://mlco2.github.io/impact/#compute
>
> **ESD definition.**
>
> We will revise the Introduction. See the blue text in the footnote on page 2.
>
> **Figure 2 is unclear.**
>
> Yes, we meant only for the black stars. We will revise the wording to clarify this point. See the blue text in Section 3.2.1.
>
> **Scale metrics are perfectly anticorrelated with model quality.**
>
> Note that this strong negative correlation only holds in this one particular scenario. In other scenarios, such as in Jiang et al. (2019) and Dziugaite et al. (2020), the correlation is strong in the other direction. Broadly speaking, if a particular theory says that a quantity should go up with model quality, and it goes down sometimes instead, then the theory is wrong, or at least has some fundamental flaw, regardless of how strong the correlation is. A prominent claim in the paper is that the correlation between test error and the generalization gap can be reversed, and we don’t know when this transition happens. If one can predict this transition, then one can use scale metrics. However, we can still use the shape metrics which don’t appear to suffer from this problem.

---

### Official Review · Reviewer_zKbc · 2022-10-23

**Confidence:** 4
**Correctness:** 4
**Technical Novelty And Significance:** 2
**Empirical Novelty And Significance:** 4
**Recommendation:** 6

**Clarity, Quality, Novelty And Reproducibility:**

**Clarity**

It is impressive that this paper is stated in such a clear way that I feel easy to absorb quite a few information it wants to deliver.  In particular it gives a good literature survey which lays down the overall background.

I only have one question which puzzled me: I cannot figure out why *Then even if we have i) access to both models’ training errors, and ii) a metric which is guaranteed to perfectly rank correlate with the generalization gap, then we still cannot determine which model as smaller test error.*

**Quality**

The study here is quite solid. For example different settings of NLP experiments are studied like across different well trained models, across different hyper-parameters and across different training phases/time. I believe it could be very useful for NLP researchers who care a lot about real-world performances.

**Novelty**

While this paper proposes several new metrics to study, its technical novelty seems not comparable with several previous works. On the other hand there is a certain degree of novelty in terms of problem itself, that is driving some data free metric to predict test set performances in NLP which would be very useful in real-world.

**Reproducibility**

Good.



**Strength And Weaknesses:**

**Strengths**

1. The problem at hand is indeed meaningful: getting the full-set dev set performance for real NLP tasks, especially that web-scale ones, are very time and resource consuming, hence having a data-free surrogate metrics for model selection would be quite useful, especially for real industry use cases;

2. The study here is quite solid which delivers convincing insights for the community.

3. The paper is generally written in a clear way.

**Weaknesses**

1. The raw innovation of this paper seems limited given its natural of being an empirical study focused work.

**Summary Of The Paper:**

This paper provides a very comprehensive empirical analysis of predicting the test set performance on NLP tasks, and reveals several interesting properties like shape based metrics are generally better than scale based ones, and some fitting strategies (for empirical spectral densities of a weight matric) like E-TPL could alleviate the issues of other typical fitting methods like pure PL based.

**Summary Of The Review:**

As stated above, despite that lacking of technical novelty, this paper is generally good given its solid empirical study which could be of good/common interest to NLP researchers.

---

> ### Author Response · Authors · 2022-11-11
> **Response**
>
> **The raw innovation of this paper seems limited.**
>
> We disagree that any empirical study is inherently of limited novelty. Indeed, we believe that our study may be of interest both to practitioners, who may use the metrics we evaluate, as well as to theoreticians, among whom the distinction between studying model quality (i.e. test error) and the generalization gap is not often discussed. In our opinion, identifying NLP as an area where this distinction is critical alone represents a important contribution, especially in light of previous similar studies which focused on CV (where the distinction is less relevant).
>
> **One question on rank correlation.**
>
> Note that the correlations reported here and in other related works are _rank_ correlations, meaning in particular that they are sensitive to shifts in two quantities being correlated.
>
> In our case, this means that the rank correlation between TestError - TrainError and Metric can be very different than the rank correlation between TestError and TrainError + Metric, specifically because the generalization metrics are almost always very large numerically in practice. To illustrate this, consider the following hypothetical example:
>
> -------------------------------------------
> Model &nbsp;&nbsp;&nbsp;&nbsp;&nbsp; |  TrainError |  TestError &nbsp;  |  Metric   &nbsp;&nbsp;&nbsp;&nbsp;&nbsp;&nbsp;&nbsp;&nbsp;&nbsp;    |
> -------------------------------------------
> Model1 &nbsp;&nbsp;&nbsp;&nbsp;|  0.03  &nbsp;&nbsp;&nbsp;&nbsp;&nbsp;&nbsp;&nbsp;&nbsp;&nbsp;        |   0.07  &nbsp;&nbsp;&nbsp;&nbsp;&nbsp;&nbsp;&nbsp;&nbsp;&nbsp;       | 150,000.0 &nbsp; |
> -------------------------------------------
> Model2 &nbsp;&nbsp;&nbsp;|  0.09  &nbsp;&nbsp;&nbsp;&nbsp;&nbsp;&nbsp;&nbsp;&nbsp;&nbsp;&nbsp;        |   0.10 &nbsp;&nbsp;&nbsp;&nbsp;&nbsp;&nbsp;&nbsp;&nbsp;&nbsp;        | 45,000.0 &nbsp;&nbsp;&nbsp;  |
> -------------------------------------------
>
> In this case, have that for model 1, TestError - TrainError is 0.04, while for model 2, TestError - TrainError is 0.01. Since the generalization metric for model 2 is smaller than that for model 1, the rank correlation between the generalization gap and the metric is equal to 1. On the other hand, if we add the train error to both sides, we have for model 1 TestError = 0.07 and TrainError + Metric = 150,000.03 and for model 2 TestError = 0.03 and TrainError + Metric = 45,000.09. Now, because of the difference in scales, the rank correlation between TestError and TrainError + Metric is $-1$, i.e. completely flipped.
> This example, while made up, is qualitatively exactly the situation that arises in actual examples, e.g. the difference between Figures 4(a) and 4(b).

---

### Official Review · Reviewer_j1Vx · 2022-10-24

**Confidence:** 2
**Correctness:** 2
**Technical Novelty And Significance:** 2
**Empirical Novelty And Significance:** 2
**Recommendation:** 5

**Clarity, Quality, Novelty And Reproducibility:**

Transferring the metric from CV to machine translation. The novelty looks a bit limited, but experiments are solid.

**Strength And Weaknesses:**

Strength:
1. The authors have done widely exploration of generalization metrics on machine translation tasks.
2. The metric can be used for machine translation model evaluation.

Weakness:
1. The paper over-claimed the contribution. The authors only work on machine translation task but claim "Evaluating natural language processing models " in the title. I would suggest authors work on GLUE/SuperGLUE/AdvGLUE dataset to further explore the robustness.
2. Not sure why the authors want to emphasize "do not need access to any training or testing data". It is ambiguous. Some scores like BERTScore (Evaluating Text Generation with BERT), BARTScore (BARTScore: Evaluating Generated Text as Text Generation) also don't need. Can the metric evaluate model only based on the model initialization?

**Summary Of The Paper:**

The authors provide the first systematic empirical study on various generalization metrics in NLP, including 400 transformers trained with varying hyperparameters. The authors measure the correlation between 28 generalization metrics and the model quality. The authors find that shape metrics consistently perform better than scale metrics for predicting model quality.



**Summary Of The Review:**

Overall, I think the authors over-claimed too much on the contribution. I would suggest the authors either add more NLP tasks in the experiments or change the title. I appreciate the large-scale experiments on machine translation tasks, while the metric mainly comes from CV and the novelty is little bit limited.

---

> ### Author Response · Authors · 2022-11-11
> **Response**
>
> **The authors only work on machine translation.**
>
> See the comment on new experiments.
>
> **BERTScore and BARTScore**
>
> BERTScore evaluates the quality of the generated text with the help of a BERT model. For a reference sentence y and a candidate sentence x, the method first uses BERT to extract token embeddings of both y and x. Then, it calculates the pairwise cosine similarity between the two sequences of embeddings and uses the maximum similarity to represent “occurrence” similar to the BLUE score. Then, importance weighting is optionally used to weigh the similarity scores.
>
> BARTScore, on the other hand, treats the text quality evaluation as a text generation task with the help of the pretrained language model BART. This method calculates the quality score by measuring the log conditional probabilities from reference or source sentences to hypothesis sentences or vice versa.
>
> Our work is different because we do not need any data and do model selection using the weight matrices only. BERTScore and BARTScore evaluate the text quality, and thus they need source or reference texts generated by humans. These metrics can serve as alternatives to BLEU, which is viewed as ground truth in our work. We updated our paper to include this discussion. See the blue text in the footnote on page 7.
>
> If we view pretrained weights as “initialization”, then yes, we can evaluate models only based on model initialization. We cannot evaluate a model using randomly initialized weight matrices because these weight values mostly contain noise.
>
> **Novelty of our paper**
>
> The main focus of this paper is not on applying generalization metrics to NLP, but on studying the many subtle issues when dealing with NLP models, such as not training to zero error. When models are trained to zero error, predicting test error is the same as predicting the generalization gap, and thus prior work often treats these two quantities equally. However, we show that for NLP, these two quantities are fundamentally different, and thus we need to rethink prior results that primarily focus on CV. Identifying NLP as an area where the difference between these two quantities is critical alone represents an important contribution.

---

### Official Review · Reviewer_rJjc · 2022-11-03

**Confidence:** 3
**Correctness:** 2
**Technical Novelty And Significance:** 2
**Empirical Novelty And Significance:** 2
**Recommendation:** 3

**Clarity, Quality, Novelty And Reproducibility:**

*Quality*: The paper conducts an extensive evaluation of generalization metrics, but I have some doubts about the choice of evaluation and the interpretation of the conclusions. In particular, the first set of evaluations (figures 2 and 3) only only compares shape metrics; the second set of evaluations (figure 4) may indicate that scale-based metrics are actually preferable (see above); and final set of evaluations (figure 5) may have limited relevance to the paper, because it is comparing correlation with model hyperparameters rather than a quality score.

*Clarity/reproducibility*: The paper is clearly written for the most part, but I have methodological questions about how the scores from different weight matrices are combined to calculate the PL metrics.

*Novelty*: To my knowledge, evaluating generalization metrics for NLP has not been addressed in prior work, and this is a valuable new contribution.

**Strength And Weaknesses:**

**Strengths**
- This paper extends the study of generalization metrics to NLP. This is a valuable contribution, because unlike in computer vision, NLP models are not typically trained to zero training error. As the authors note, even if a generalization metric is rank-correlated with the generalization gap, it may not be correlated with the test error due to differences in training error. There are also differences in neural network architecture (transformers vs. CNNs).
- The paper considers a fairly wide range of metrics and hyperparameters.

**Weaknesses**
- There are two main components in the paper (1) Evaluating natural language processing models with generalization metrics, and (2) advocating the use of a certain class of generalization metrics based on heavy-tail self-regularization theory (HT-SR), including several new metrics based on fitting exponentially truncated power laws to the ESD of the weight matrix. In my opinion, the second point is largely orthogonal to the first, and it's difficult to assess whether these HT-SR metrics are preferable to existing metrics due to methodological differences with prior work [1, 2]. I think the paper would be stronger if it placed less emphasis on shape metrics and focused more on evaluating generalization metrics for NLP, using a similar evaluation methodology to prior work.
- Interpretation of Figure 4 (comparing generalization metrics): Figure 4 indicates that scale-based metrics are positively correlated with generalization gap but have a strong negative correlation with model quality, meaning a lower value of the generalization metric is associated with higher value of model quality. Isn’t this to be expected? We would expect that a low generalization gap is correlated with a high quality score. From this point of view, the metrics at the bottom of Figure 4a seem better. The other evaluations in the main paper only compare shape metrics, or compare correlations with hyperparameters rather than quality (see next point), so overall I am not convinced that shape metrics are preferable to other metrics.
- The evaluation setting in 3.3 is puzzling to me. From what I can understand, this section does not measure the correlation between generalization metrics and model quality, but instead measures correlation with architecture parameters like depth and hidden dimension. It's not clear how this section supports the argument of the paper. This section might be more convincing if it compared models that were trained on the same data and then evaluated on a consistent quality metric.
- Justification for HT-SR metrics: In particular, is there any theoretical justification for these metrics? The authors propose several different shape metrics (based on PLs and E-TPLs, different parameters, and goodness-of-fit metrics). Which of these should be preferred?
- Methodological differences from prior work: Prior work [1, 2] has explored methodological questions about comparing generalization measures and proposed metrics and methodologies aimed at controlling for noise and unwanted correlations---in particular using Kendall's tau correlation and controlling for Monte Carlo noise across environments. This paper reports Spearman's rank correlation and does not explicitly address the questions raised by [1, 2] about how to robustly evaluate generalization metrics.
- "Generalization metrics that do not need access to any training or testing data": The authors argue that shape metrics are preferable "without access to training or testing data". It would be valuable to be more clear about how existing metrics depend on the training data: to my understanding, metrics based on generalization bounds depend only the number of samples in the training data, which is a relatively mild requirement (you need to know the size of the data, but don't need access to the data itself). Also, none of the other generalization metrics considered use the testing data, so that should not be considered a special property of shape metrics.
- The authors note that "NLP pretraining datasets are typically web-scale and are challenging to access", but their evaluation is focused on a machine translation dataset (where the data is available), so the argument about NLP pretraining is not entirely relevant.
- Correlation evaluated on optimally trained models: It's not clear to me that this is a reasonable evaluation setting. How are you supposed to determine which model is optimally trained without access to testing data? If you can determine whether a model is optimally trained, why do you need a generalization metric?

- The paper only considers one dataset and quality metric (BLEU score). It would be valuable to consider a wider range of commonly used NLP benchmarks, such as the GLUE benchmark.
- It's not entirely clear to me which weight matrices are being considered in these metrics, and how the scores of different weight matrices are aggregated. More generally, it would be valuable to discuss another difference from prior work, which is that the models considered here are based on the transformer architecture, while prior work has mainly studied CNNs (and most theoretical metrics have been developed for feed-forward networks, but see [3].)

[1] Jiang et al., 2019. Fantastic generalization measures and where to find them.
[2] Dziugaite et al., 2020. In search of robust measures of generalization.
[3] Long and Sedghi, 2019. Size-free generalization bounds for convolutional neural networks.

**Summary Of The Paper:**

This paper empirically measures correlations between various generalization metrics and model quality (BLEU score) on a machine translation dataset across a range of hyperparameters. Compare to prior work, the main differences are:
- Focusing on NLP, rather than computer vision
- Focusing on metrics that predict test error rather than generalization error
- Focusing on a class of metrics based on whether the empirical spectral density (ESD) of the weight matrix follows a power law.

**Summary Of The Review:**

My recommendation is to reject this paper. The paper addresses an interesting problem, evaluating generalization metrics for NLP, but it is difficult to interpret the results due to the secondary focus of the paper on advocating the use of a certain class of metrics, and I find that some of the analysis does not seem to support the main claims of the paper. I think the paper would be stronger if it placed less emphasis on shape metrics and focused more on evaluating generalization metrics for NLP, using a similar evaluation methodology to prior work.

---

> ### Author Response · Authors · 2022-11-11
> **Response**
>
> **Two main components in the paper. Less emphasis on shape metrics.**
>
> The main argument of this paper is that generalization measures should be able to predict test error instead of the generalization gap. We base our study on Transformer models in NLP. But the difference is not NLP per se, but the many subtle issues when dealing with NLP models, such as not training to zero error. Our evaluation framework is similar to that of [1,2], e.g, in that we also study high-dimensional hyperparameter grids on a benchmark dataset. In addition to results similar to [1,2] (Figures 10 and 11 in the supplementary material), we also propose our own evaluation frameworks, such as evaluating tuned models (Figure 2), evaluating time-wise correlations (Figure 3) and evaluating pretrained Huggingface models (Figure 5).
>
> Advocating the use of heavy-tail self-regularization theory is motivated by our paper’s results on the NLP models. These results on NLP extend prior literature’s results on CV models (Mahoney & Martin 2021b; Martin et al., 2021), showing that generalization metrics motivated by HT-SR work on a wide range of models. We hope the reviewer can re-evaluate our paper based on the answers to this and other questions.
>
> **Interpretation of Figure 4.**
>
> Please note that, in Figure 4a, we have signed each of the metrics to ensure that a positive correlation corresponds with the expected trend, e.g. for a scale metric, a positive correlation is associated with the claim that smaller metrics imply better model quality, coming from the general rule of thumb of training a classifier to 0% error. So in Figure 4a, what actually occurs is that larger scale metrics correspond with better model quality, contrary to what is expected. Again, this is because our models are not trained to 0% error, and so the scale metrics are correlating with training error in a nontrivial way. We have presented our results in this way to keep consistency in what the correlations represent. This was discussed in detail in the supplementary material due to space constraints, but we will move this back into the main paper to avoid confusion. See the blue text on Page 8.
>
> Similarly, in Figure 4b, we ensure that a positive correlation corresponds with the expected trend, i.e., when the generalization gap is smaller, the metric is smaller.
>
> **Evaluation setting in 3.3.**
>
> Note that we report the correlation between generalization metrics and size parameters because, for all our evaluated Huggingface model series, larger models have universally better downstream performance. Also, note that “comparing models that were trained on the same data and then evaluated on a consistent quality metric” is done for all the other experiments on the WMT dataset.
>
> See the following references.
>
> **BERT series**
> https://arxiv.org/pdf/1908.08962.pdf
> Figure 7: Comparison against analysis baselines.
>
> **GPT2 series**
> https://cdn.openai.com/better-language-models/language_models_are_unsupervised_multitask_learners.pdf
> Figure 1: Zero-shot task performance of WebText LMs as a function of model size on many NLP tasks.
>
> **ALBERTv1 and ALBERTv2 series**
> https://github.com/google-research/ALBERT
> See the Table on the performance of ALBERT V1 and V2 on different downstream tasks.
>
> **T5**
> https://arxiv.org/pdf/1910.10683.pdf
> Table 14: Performance of our T5 variants on every task we study.
>
> **DialoGPT**
> https://arxiv.org/pdf/1911.00536.pdf
> Table 3: 6K Reddit multi-reference evaluation.
>
> **FlauBERT**
> https://arxiv.org/pdf/1912.05372.pdf
> Table3: Accuracy on the CLS dataset for French.
> Note: There are many tables in this paper. Only results for FlauBERT-base and FlauBERT-large are reported, and the results for FlauBERT-small are missing. However, the authors mentioned in their github that “flaubert-small-cased is partially trained so performance is not guaranteed. Consider using it for debugging purpose only.”
> GitHub code link:
> https://github.com/getalp/Flaubert
>
> **Funnel Transformer**
> https://github.com/laiguokun/Funnel-Transformer
> The authors show that larger Funnel models work better on four different downstream tasks.

---

> > ### Author Response · Authors · 2022-11-11
> > **Response**
> >
> > **Theoretical justification for HT-SR metrics.**
> >
> > If the reviewer is asking for generalization bounds, there aren’t any that would specifically identify the metrics we present in this paper. The shape metrics we use are principled ways of measuring heavily-tailed structure in weight matrices’ ESDs, which is hypothesized to be indicative of model performance. This hypothesis is justified in several prior works on the HT-SR approach, e.g. Mahoney & Martin 2021b, published in JMLR and other top venues. One of our objectives was to explore how well these metrics (and a few variants) actually perform in NLP tasks. Given that they appear to work surprisingly well, we believe that the empirical results presented here motivate additional theory as an important next step.
> >
> > Our goal is not to suggest one robust metric, but to test many and identify when they work well and when they don’t. Nevertheless, our experimental results do seem to suggest that the goodness-of-fit metrics generally perform the best. Our central takeaway is that shape metrics uniformly outperform scale metrics if the goal is to predict test error instead of the generalization gap (otherwise, for generalization gap, scale metrics still appear to perform very well).
> >
> > **Methodological differences from prior work.**
> >
> > We now report the results using Kendall’s tau correlation; see Figure 14 in the supplementary materials. As one would expect, the results are effectively the same. Controlling for Monte Carlo noise as in [2] -- by rejecting model pairs whose difference in error (or error gap) is less than a threshold -- indeed would be ideal. However, the models we train here are considerably larger and more expensive to train than the much smaller networks considered in [2], making rejecting model pairs impractical. Still, to get a sense of the variability in rank correlation (perhaps due in part to Monte Carlo noise), we can refer to the ranges reported in, e.g., Figure 4.
> >
> > **Generalization metrics that do not need access to any training or testing data**
> >
> > It is not true that “metrics based on generalization bounds depend only on the number of samples in the training data”. There are many generalization bounds (indeed, some of the most successful ones) that require evaluating margin and flatness scores (such as the PAC-Bayes bounds), which require training data. Data-free generalization metrics can be helpful when any reasonable dataset is hard to use or obtain, e.g., due to privacy or compliance issues. The Weightwatcher tool, which we use to evaluate the HT-SR metrics, has apparently proven to be practical in commercial settings where the data is hard to obtain. Also, our restriction to metrics which “do not need access to testing data'' is to immediately rule out test accuracy, which remains the most typical measurement of generalization performance to date.
> >
> > **NLP pretraining and machine translation.**
> >
> > We need to keep the balance between making an impact and being able to evaluate our results rigorously. Here we are commenting on the fact that these metrics can be used in SOTA situations where NLP pretraining is web-scale. For obvious reasons, it is not sensible to conduct our tests on these metrics in situations where we do not have the underlying data to compare against.
> >
> > However, note that we do also report results on Huggingface Transformers. The results on Huggingface Transformers show that we can apply our methods when the pretraining datasets are hard to access.
> >
> > **Correlation evaluated on optimally trained models.**
> >
> > Please note that in Figure 2, we are trying to grid-search the learning rate, which is the standard practice when training a model. The setting we are interested in here is transfer learning, where the practitioners are not the ones who are training the model, but still require some more trustworthy form of model selection criteria to assess which model should be used. It seems reasonable to expect that published pretrained models have at least been competently trained. Using Figure 2, we want to emphasize that one could create simple anti-correlations by making a model less well-trained. However, evaluating only well-trained models is more impactful in practice.

---

> > > ### Author Response · Authors · 2022-11-11
> > > **Response**
> > >
> > > **One dataset.**
> > >
> > > See the comment on new experiments.
> > >
> > >
> > > **Details on weight matrices.**
> > >
> > > Thanks for pointing out the reference. We will include it in our updated version. See the blue text in the conclusion section. As the reviewer has pointed out, we mainly focus on the Transformer architecture, which is different from prior work, especially CNNs (for which the spectral norm of the 4D kernels have to be evaluated differently). In this work, we do not differentiate between different Transformer layers, and we aggregate the layer-wise metrics by simple averaging. Also, when calculating the ESDs of the weight matrices, we treat the query, key and value matrices as separate weight matrices. That being said, we do benefit from large Transformer blocks, because the increased number of eigenvalues of a large weight matrix typically leads to better powerlaw-fitting quality. Further exploration of the different roles of these matrices can potentially lead to improved quality prediction.

---

> ### Comment · Reviewer_rJjc · 2022-12-09
> **Response to the authors**
>
> Thank you to the authors for their response and updates to the paper, and apologies for the late response. In particular, thanks for correcting my understanding of Figure 4 and for the new results in appendix E. However, the changes do not substantially change my review and I still recommend to reject the paper, because I am still not convinced by the experimental setup.
>
> First, previous work [1, 2] has pointed out pitfalls of evaluating generalization measures based on correlations with generalization, and proposed more frameworks for how to aggregate results from across hyperparameter settings to get more  robust conclusions. I am specifically thinking of methods like [2], that compare correlations across pairs of experimental settings. This submission doesn’t adopt this framework, and as a result I find it hard to evaluate whether these metrics are truly predictive of generalization across different settings, and not just correlated with some other hyperparameter. From my understanding, Figure 4 compares correlation across training epochs, averaged over settings, instead of correlation across settings. The results in the appendix provide additional breakdowns by fixing different hyper parameters,  but it’s difficult to get an overall picture of how these metrics are correlated across all hyperparameter changes.
>
> I also still have concerns about the comprehensiveness of the evaluations in the main paper. In particular, the first set of evaluations (figures 2 and 3) only compares shape metrics, and the final set of evaluations (figure 5) compares correlation with model hyperparameters rather than a quality score. If hyperparameters like width and depth are correlated with quality, we can just use those metrics instead. The new results in appendix E compare the correlation between a single shape metric and data size, which does not substantially improve my understanding of how metrics are correlated with quality across hyperparameter settings. Therefore, I don’t think the results clearly show that shape metrics are more predictive of quality than generalization metrics.
>
> I think the paper would be stronger if the evaluation was more in line with prior work, and if offer clearer insight into how different metrics are correlated with quality across different hyperparameter changes.
>
> [1] Jiang et al., 2019. Fantastic generalization measures and where to find them. [2] Dziugaite et al., 2020. In search of robust measures of generalization.

---

> > ### Author Response · Authors · 2022-12-10
> > **Response**
> >
> > We are glad that we have clarified Fig 4. Still, we would appreciate it if you could clarify your concern about the experimental setup, especially since your score is well below the score of the other three reviewers.
> >
> > You point to [1][2], but the generalization measures studied in these two papers require data to calculate. This assumption is strong for practical model evaluation tasks. It is difficult to make our paper more in line with [1][2] without significantly degrading its novelty.
> >
> > From a cost perspective, we deal with Transformers. Examining the code in [2] shows that our base six-layer Transformer is 64.4 times larger than the four-layer 8 × 25 width NiN for Cifar10. Fully reproducing the results in [2] would be absurdly costly, and it would mean at least 10-fold increase in computational and storage costs for us due to the random seeds used for evaluating the robust sign-errors.
> >
> > The main contribution of our paper is to do something different than prior work, highlighting their fundamental limitations.
> > - We show that the generalization gap and model quality differ and can be correlated or anticorrelated. We thus focused on predicting the trends of model quality, while [1][2] focus on the generalization gap.
> > - We pointed out that evaluating models without data is crucial for NLP tasks, while [1][2] requires data.
> > - We evaluate 51 Huggingface Transformer models, while [1][2] cannot do that because of the requirement of accessing data.
> >
> > Nonetheless, we make our paper in line with [1][2] so that the reviewer can evaluate our progress.
> > - We studied the same set of metrics as [1][2] (except for some duplicates).
> > - We studied how the trends vary for different hyperparameters (sample, width, lr).
> > - We now provide results on Kendall-tau after the reviewer suggests it.
> >
> > For the questions raised in the new post, we apologize that we cannot revise our paper further due to the policy of Phase 2.
> > - **Reproducing results from [2].** The conclusion in [2] is that "no existing complexity measure has better robust sign-error than a coin flip." This means that, even if [2] considers measures that can access data, no measure works if the goal is to achieve a small robust sign-error. We thus question the necessity to evaluate the robust sign-error in [2] when our metrics do not have access to data.
> > - **Figures 2 and 3 only compare shape metrics.** We do have the results on all other measures. E.g., see Figure 7. We can provide more examples of generalization measures from prior work. However, as we show in the summarized results in Figure 10, they give the wrong predictions (e.g., see Figure 7).
> > - **Figure 5 compares the correlation with width and depth.** We have provided extensive evidence in the rebuttal that larger models from the Huggingface website have better downstream quality. Furthermore, we want to emphasize that generalization measures requiring data cannot evaluate these models.
> >
> > In summary, we provide a sufficiently novel and practical framework, and our goal is not to simply generalize prior work to NLP.

---

### Author Response · Authors · 2022-11-11
**New experiments**

Since multiple reviews mentioned new datasets, we post one comment to respond to these questions.

We provide new experiments in Appendix E.
- Roberta-base (Liu et al., 2019) trained on the masked language modeling task using the Wikitext-103 dataset (Merity et al., 2016) and finetuned on MNLI (Williams et al., 2018).
- Six-layer base Transformers trained on the language modeling task using the Wikitext-103 dataset;
- Six-layer base Transformers trained on the next-word prediction task using the Reddit dataset (Baumgartner), following the implementation in Bagdasaryan et al. (2020);

In addition, the experiments on Huggingface Transformers (Figure 5) already include the GLUE benchmark. Finetuning these Transformers on downstream tasks will generally have the same trend as the pretraining quality. However, one can indeed significantly alter the finetuning result by following non-standard practices.

---

### Author Response · Authors · 2022-11-17
**A gentle reminder that Discussion Stage 1 will end soon**

Dear reviewers, thanks for your constructive feedback. Since it is close to the end of stage 1, please tell us if we should include anything further in the revised draft. We are more than happy to clarify if anything is unclear.

---

### Decision · Program_Chairs · 2023-01-20

**Decision:**

Reject

**Justification For Why Not Higher Score:**

This is a harder decision to make because the paper seems to be well-written and has clear merits.
However, several reviewers feel that the empirical results are not convincing enough in the current draft.

**Justification For Why Not Lower Score:**

N/A

**Metareview: Summary, Strengths And Weaknesses:**

This paper provides a comprehensive empirical study of the correlations between various generalization metrics and test performance of NLP models.

This paper is a borderline paper and has been discussed between AC and reviewers.

Most reviewers appreciate the thorough studies that have been done in this paper, and the paper is well-written and condenses a lot of information of background. The experiments are also generally solid.

There were several major concerns initially:
* The results were only reported on a single dataset WMT’14, while the scope of the paper targets “Evaluating NLP models..”
* Whether the paper offers a robust, comprehensive comparison of quality measures across hyperparameters (the standard needs to be high given the paper is a purely empirical paper, and some reviewers questioned the technical novelty of the work)

The authors added new experiments on MNLI, WikiText-103, and Reddit during the rebuttal phase. However, several reviewers still believe the new results are not convincing enough (e.g., only comparing the correlation between a single shape metric and quality across different amounts of training data, whether they represent a broad range of NLP tasks).

In summary, due to the above concerns and there isn’t any (strong) support for the paper, I can’t recommend acceptance of the paper in its current form.